# Commonality Evaluation and Prediction Study of Light and Small Multi-Rotor UAVs

Yongjie Zhang , Yongqi Zeng  and Kang Cao *

Civil Aviation School, Northwestern Polytechnical University, Xi'an 710072, China; zyj19191@nwpu.edu.cn (Y.Z.); zengyq@mail.nwpu.edu.cn (Y.Z.)
* Correspondence: caokang@mail.nwpu.edu.cn

**Abstract:** Light small-sized, multi-rotor UAVs, with their notable advantages of portability, intelligence, and low cost, occupy a significant share in the civilian UAV market. To further reduce the full lifecycle cost of products, shorten development cycles, and increase market share, some manufacturers of these UAVs have adopted a series development strategy based on the concept of commonality in design. However, there is currently a lack of effective methods to quantify the commonality in UAV designs, which is key to guiding commonality design. In view of this, our study innovatively proposes a new UAV commonality evaluation model based on the basic composition of light small-sized multi-rotor UAVs and the theory of design structure matrices. Through cross-evaluations of four models, the model has been confirmed to comprehensively quantify the degree of commonality between models. To achieve commonality prediction in the early stages of multi-rotor UAV design, we constructed a commonality prediction dataset centered around the commonality evaluation model using data from typical light small-sized multi-rotor UAV models. After training this dataset with convolutional neural networks, we successfully developed an effective predictive model for the commonality of new light small-sized multi-rotor UAV models and verified the feasibility and effectiveness of this method through a case application in UAV design. The commonality evaluation and prediction models established in this study not only provide strong decision-making support for the series design and commonality design of UAV products but also offer new perspectives and tools for strategic development in this field.

**Keywords:** light and small multi-rotor UAVs; commonality evaluation; design structure matrix; convolutional neural network; commonality prediction



## 1. Introduction

As unmanned systems are leaping forward, UAVs have been confirmed as a vital branch of unmanned systems. Moreover, UAVs have become one of the most rapidly developing and attention-grabbing areas of unmanned systems. UAVs have become increasingly popular over recent years and have been extensively employed in different fields because of their unmanned, multifunctional, intelligent, and economic characteristics [1]. Existing research on UAVs has primarily investigated the technology, performance, and applications of UAVs while placing stress on UAV flight control technology [2], intelligent sensing and obstacle avoidance systems [3], sensor technology, as well as data communication [4]. However, UAVs, a type of industrial product, are capable of creating economic benefits to contribute to social and economic development, and they have huge development potential. It is noteworthy that in the field of civil UAVs, the industrial upgrading of the industry and social product development can be boosted by the broad application of UAV products. According to the latest "General Aviation Industry Development White Paper (2022)" released by the Aviation Industry Corporation of China, the global civil UAV market size has exceeded CNY 160 billion in 2021, and this amount will surge to CNY 500 billion in 2025. The demand for UAVs is increasingly high due to the continuously expanding downstream

applications of UAVs, such that the global civil UAV market will achieve inevitable rapid growth in the future.

The level of technology research and development performed by UAV companies takes on a critical significance to the size of the market share, and continuous investment in research and development is required to continuously optimize UAV systems, technologies, and solutions. Otherwise, it will be challenging to gain a foothold in the market. Market-driven multi-rotor UAV industrial product series development has served as an essential development strategy for major UAV companies. The concept of commonality is critical to the research on serialized products. Commonality refers to a series of asset reuse and sharing methods developed in accordance with generalized similarity [5]. Commonality design covers the ideas of inheritance, standardization, and modularity while considering innovative needs; it serves as a market-driven "win-win" implementation method for enterprises/users. The commonalty design primarily aims to reduce costs and increase efficiency. With the use of commonality design in the development strategy of serialized light and small multi-rotor UAV products, the development time of UAVs can be shortened, the production efficiency of enterprises will be increased, product costs will be lowered, product launch will be expedited, and more market share of UAVs can be captured. For instance, DJI, the world's most famous UAV company, has Mavic-series UAVs and Phantom-series UAVs, in addition to others, thus markedly contributing to DJI's domination of the global consumer UAV market segment.

For commonality, extensive systematic studies have been conducted on commonality design and commonality evaluation. Natarajan et al. [6] have suggested that using common components can significantly reduce product design and manufacturing time, and the commonality between different components should be determined, which is critical to shortening the cycle time of novel product design. Blackenfelt [7] redesigned a range of lifting tables to determine the right balance between commonality and diversity and between low cost and unique product performance to maximize profits. Additionally, noise analysis was conducted to obtain universally designed components. Fujita et al. [8] formulated the design problem of generic components as an optimization problem and addressed the common design problem of a car opening and closing tiller using a genetic algorithm. Hölttä-Otto [9] investigated the design of product diversity in a modular architecture and the design of modular commonality in a computation-oriented method such that the product variety design problem was systematically summarized as a 0–1 integer programming problem. Kim et al. [10] proposed a step-by-step method to determine the optimal sustainable product series architecture design, used to balance product commonality and the protection of intellectual property rights for sensitive components.

In the field of aviation, Nuffort [11] identified the levels of commonality in an aircraft family of products through the complete life cycle analysis of the aircraft and analyzed the benefits of commonality over the product life cycle from the perspectives of the manufacturer and the operator, respectively. Bador [12] studied the design of cockpit commonality in civil aircraft and then divided commonality into temporal vertical commonality and horizontal commonality. He analyzed the measurement result of the same characteristics in the product family in accordance with standardization, modularity, and reusability such that the design of cockpit commonality in civil aircraft can be guided. Using two types of civil aircraft of an aviation manufacturer as an example, Xi et al. [13] classified the design differences and commonalities in terms of the functions and architecture, working principles, and main components of the air management system. Cai et al. [14] analyzed and categorized the cockpit design elements of civil aircraft and built a database of cockpit commonality design elements of civil aircraft. To be specific, the database covers nearly 1100 items of cockpit commonality design elements of five models (e.g., B737-800 and A350). Zhang et al. [15] built the commonality system of civil aircraft operation support at two levels (i.e., product level and business process level) in accordance with the characteristics of Chinese civil aircraft operation support at this stage. Carlos et al. [16] mentioned that in the design of jet aircraft series, original equipment manufacturers (OEMs) can reduce

development and manufacturing costs through family concept design, although there is a trade-off between member performance and commonality.

Based on different parameters (e.g., number of identical parts, connection method, and cost), Thevenot et al. [17] summarized commonality indexes, including DCI for evaluating commonality from the component perspective and CMC [18], which considers commonality in product components, size, shape, material, cost, and so forth. Zhang et al. [19] proposed a serialized aircraft commonality index in accordance with the component decomposition hierarchy and classified the serialized civil aircraft commonality index into two categories (i.e., component commonality index and cockpit commonality index). The former has a major function of macroscopically evaluating the degree of commonality among aircraft components and the whole aircraft in the aircraft series. The latter is capable of evaluating the degree of commonality among aircraft cockpits. Moreover, Zhang et al. evaluated the commonality using analytic hierarchy process and the fuzzy comprehensive evaluation method by building a commonality evaluation system for civil aircraft maintenance technical publications [5]. To evaluate the commonality degree of the commonality indexes, intelligent optimization algorithms can be combined to evaluate the commonality of the products. Chowdhury et al. [20] developed a comprehensive product platform planning (CP3) method that proposes a matrix-based measure of commonality, applying the mixed discrete particle swarm optimization (MDPSO) algorithm to measure the degree of commonality. Thevenot et al. [21] have developed a method to evaluate the degree of commonality in a product family using genetic algorithms and commonality indexes to weigh the commonality and uniqueness between products for the design or redesign of a product family. Takai [22] represented the commonality design of a series of products in matrix form and incorporated it as a factor in cost calculation, thus enabling the calculation of the impact of commonality design on costs. Zhang et al. [23] summarized the concepts and calculation indicators related to commonality.

From the analysis of the above studies, it is evident that although the commonality research of civilian aircraft has attracted significant attention among industry researchers, the study of commonality in multi-rotor UAVs is relatively scarce, especially in terms of commonality evaluation. Existing evaluation methods are mostly based on qualitative approaches or rely on a limited number of parameters to establish simple algorithms. These methods, while universal for various serialized products, are limited in accurately assessing specific product features. Therefore, inspired by a series of studies on intelligent design optimization algorithms for UAVs, this paper proposes a new methodology. For instance, Ganesan [24] proposed the bionic optimization leader election (BOLD) scheme for predicting UAV lifespan, yielding more accurate results than traditional methods. Li [25] developed a rapid evaluation method for assessing the endurance of agricultural UAVs and verified the accuracy of the method through testing. Zhang et al. [26] developed the LASSA-RRT algorithm to enhance the global search advantage in UAV trajectory planning. Li [27] suggested an adaptive control scheme based on a fixed-time observer (FTOAC) for UAV tracking control. Additionally, there are numerous UAV design optimization studies utilizing intelligent algorithms [28,29]. Inspired by these studies, this paper introduces a commonality quantification evaluation method based on the UAV design structure matrix and a commonality prediction method based on convolutional neural networks. This allows UAV manufacturers to have a quantitative understanding of the commonality between their models and market models in the early design stages, providing data support for product positioning and commonality design. The main work and innovations of this study include:

(1) Establishing indicators for quantitative evaluation of commonality among existing light small-sized, multi-rotor UAV models based on their feature variables.

(2) Calculating the contribution of each component in the UAV product design structure matrix to represent its importance in the system, serving as the basis for calculating indicator weights, thereby constructing a commonality evaluation model for light small-sized, multi-rotor UAVs.

(3) Collecting characteristic data of typical light small-sized multi-rotor UAVs and constructing a commonality dataset using the commonality evaluation model. Furthermore, by applying a convolutional neural network algorithm and training with this dataset, a commonality prediction model for light small-sized multi-rotor UAVs is established, aiming to achieve the goal of predicting commonality based on a small amount of feature data.

The narrative structure of this paper is illustrated in Figure 1.

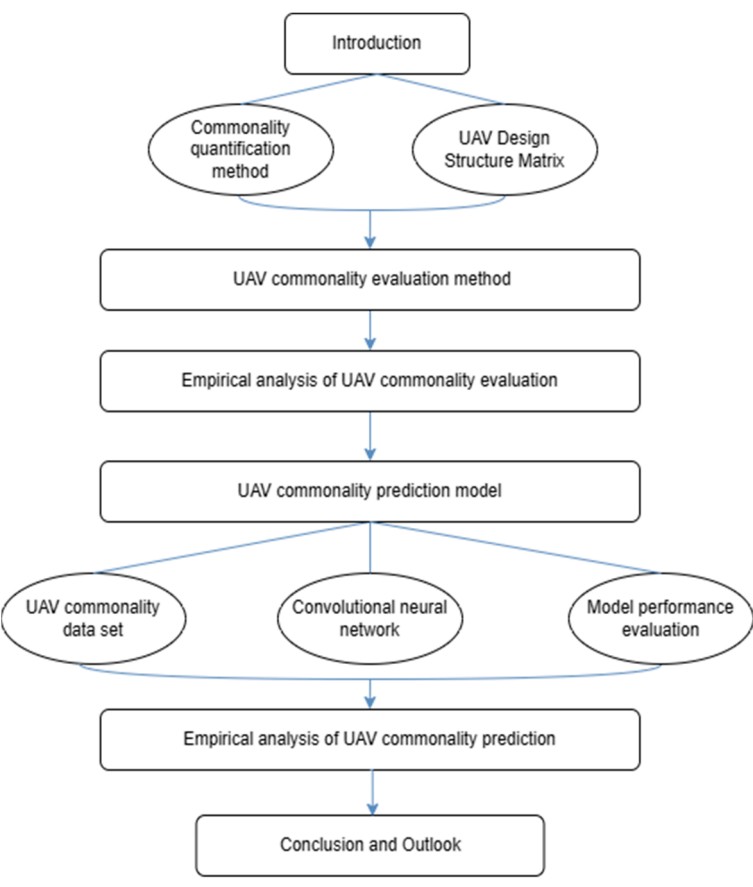

**Figure 1.** The narrative structure flowchart of the article.

## 2. Commonality Quantification Method

Multi-level commonality evaluation indexes should be built to evaluate the commonality of complex products. In general, a set of comparison characteristics data is available for the commonality indexes exhibiting quantitative evaluation characteristics. If the commonality index exhibits only one evaluation feature, the comparison data can be a set of individual numbers. The comparison data comprise a set of vectors if there are multiple evaluation features of the commonality index. The similarity and the distance between the data can be quantified to represent the commonality of the indexes. The data of the commonality indexes of a set of comparison samples are expressed as $X = (x_1, x_2, \ldots x_m)$ and $Y = (y_1, y_2 \ldots, y_m)$, where m denotes the dimension of the vector ($m \geq 1$), and $x_i$ and $y_i$ respectively represent the quantified values of the $i$th feature of the two samples. Several typical distance measures and similarity calculation methods are illustrated in the following.

(1) Minkowski distance.

$$d_{XY} = \sqrt[p]{\sum_{i=1}^{m} |x_i - y_i|^p} \tag{1}$$

The Minkowski distance refers to a generalized expression of the distance metric, and the most appropriate distance metric can be determined using the value of $p$. When $p = 1$,

$d_{XY}$ denotes the Manhattan distance; when $p = 2$, $d_{XY}$ represents the Euclidean distance; when $p = \infty$, $d_{XY}$ expresses the Chebyshev distance. The larger the distance obtained after calculation by the distance metric, the lower the commonality of the indexes indicated will be and vice versa.

The Minkowski distance can represent different distance measures, which are dependent on the $p$-value but are subjected to the same drawbacks (e.g., reliance on feature units when they face problems in high-dimensional spaces). Moreover, the flexibility of $p$-values may become a drawback under complex problems that require extensive calculations to determine the appropriate $p$-value. Since the number of features evaluated for the commonality index is small and the dimensionality of the vector is low when the commonality evaluation of light and small multi-rotor UAVs is being performed, the $p$-value is taken as 2.

(2) Cosine similarity.

$$s_{XY} = \cos\theta_{XY} = \frac{\sum_{i=1}^{m} x_i y_i}{\sqrt{\sum_{i=1}^{m} x_i^2} \times \sqrt{\sum_{i=1}^{m} y_i^2}} \tag{2}$$

Cosine similarity refers to a measure of direction whose magnitude is determined by the cosine between two vectors, whereas the magnitude of the vectors is ignored. Cosine similarity is generally employed for high-dimensional vectors for which the numerical size does not take on any significance. The main disadvantage of cosine distance is that it only considers the direction of the vectors, instead considering the numerical size.

During the evaluation of the commonality with cosine similarity, since the range of values $s_{XY}$ is $[-1,1]$ and there is no negative commonality, it is specified that the commonality is 0 when $s_{XY} < 0$.

As depicted in Figure 2, if the Minkowski distance metric only quantifies the commonality, only the distance $D$ shown in the figure can be obtained, while the effect of the direction is ignored. At the same time, the distance range is $[0,\infty]$, while the value of the commonality basically ranges from 0 to 1, thus hindering the representation of the commonality after quantification. It is assumed that the commonality is quantified using cosine similarity such that only $\theta$ can be obtained (Figure 2), which can only represent the difference in direction and ignore the effect of distance.

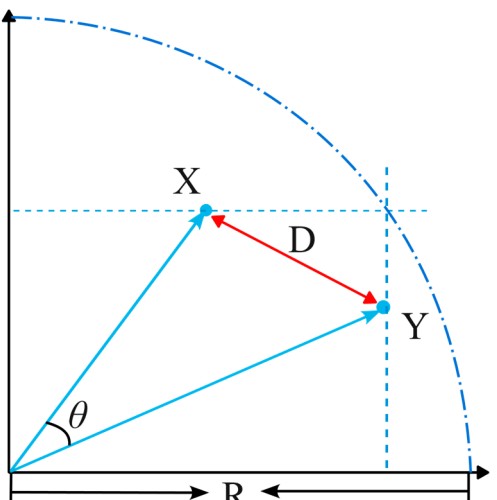

**Figure 2.** Commonality quantification schematic.

Thus, a new vector $Z = (\max(x_1, y_1), \max(x_2, y_2), \ldots, \max(x_m, y_m))$ is formed by taking the maximum value of the respective dimension of $X$ and $Y$ to map the Minkowski distance to the $[0,1]$. $R$ in Figure 2 represents the modulus of the new vector $Z$ ($R = \|Z\|$), and the maximum possible distance between $X$ and $Y$, such that $R$ is considered the

maximum distance, as expressed in Equation (3). Subsequently, the ratio of the actual and maximum distances is adopted to map Minkowski distances in the range [0,1].

$$d_{max} = R = \sqrt{\sum_{i=1}^{m}\left(\max\left\{x_i, y_i\right\}\right)^2} \tag{3}$$

In brief, Minkowski distance and cosine similarity are combined to propose a new method for numerical quantification of the commonality index. The specific equations expressed in Equations (4) and (5) are obtained by combining Equations (1)–(3).

$$CI = \begin{cases} w_1\left(1 - \frac{d_{XY}}{d_{max}}\right) + w_2 s_{xx} & s_{XY} \geq 0 \\ 0, & s_{XY} < 0 \end{cases} \tag{4}$$

$$w_1 + w_2 = 1 \tag{5}$$

where $CI$ denotes the value of commonality; $d_{XY}$ represents the Minkowski distance; $s_{XY}$ expresses the cosine similarity; $w_1$ and $w_2$ are the weights of Minkowski distance and cosine similarity, respectively, and the sum of both is 1. Since the distance and direction between the comparison data take on equal importance in evaluating commonality, $w_1$ and $w_2$ are taken as 0.5.

However, a unique form exists in vectors $X$ and $Y$, i.e., $x_i$ and $y_i$ take values of only 0 and 1, respectively. These values represent the existence of the evaluation features for the commonality index, and $X$ and $Y$ can be represented as a 0–1-encoded data structure. For such commonality indexes, the Hamming-distance-based metric is a simpler and more convenient measure for quantifying the commonality of indexes, which is also consistent with the most basic commonality theory. As depicted in Figure 3, the Hamming distance of 2 indicates the existence of two different characteristics which also do not conform to the interval requirement of the commonality value, and the commonality cannot be quantified using Hamming distance alone. The same idea as mapping the Minkowski distance can be implemented, in which the ratio of the Hamming distance and the maximum distance is adopted to complete the mapping. For instance, the OR operation in the logical operation for the data represented by $X$ and $Y$ in Figure 3 yields 11,011, which means that there a total of four features of $X$ and $Y$ exist. Furthermore, the possible maximum distance between $X$ and $Y$ is 4. Subsequently, the commonality of $X$ and $Y$ is calculated as: $1 - 2/4 = 0.5$.

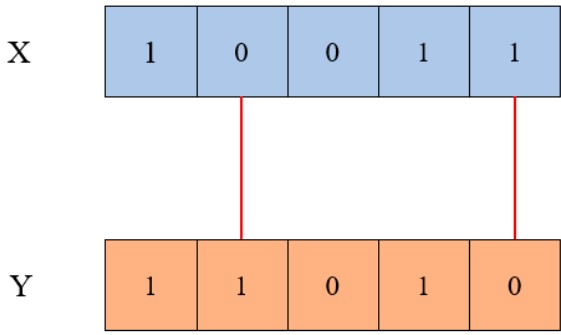

**Figure 3.** Hamming Distance.

Accordingly, the commonality indexes of 0–1 encoded data structures can be quantified using the Hamming-distance-based method for commonality, which is written as:

$$CI = 1 - \frac{\sum X \oplus Y}{\sum X \vee Y} \tag{6}$$

where $\oplus$ denotes the IOS-OR operation, indicating how the Hamming distance is calculated; $\vee$ represents the OR operation; and $\sum$ expresses the summation of a series (e.g., the series $\sum 11011 = 4$).

## 3. Light and Small Multi-Rotor UAV Commonality Evaluation Model

### 3.1. Light and Small Multi-Rotor UAV Product Breakdown Structure

The product breakdown structure (PBS) is effective in elucidating the physical components of a particular product or system. The Product Breakdown Structure is similar to the Work Breakdown Structure, and it is adopted to simplify a complex project or product into manageable parts. As a result, the team can gain more insights into the product composition and suggest conditions that the component design should conform to. The formal PBS originates from a hierarchy, which essentially breaks down the product into the required components. The above-described breakdown aims at providing a visual representation of the product components and their correlations such that product planners are enabled to obtain a visual representation of the product composition and more insights into the requirements and functionality of the final product.

In accordance with the composition of the light and small multi-rotor UAV, the three-level decomposition structure is obtained after the product structure decomposition of the UAV (Figure 4). The UAV components in the third level of the UAV decomposition structure comprise the base elements for the development of the UAV design Sstructure matrix. The UAV design structure matrix serves as an essential tool to determine the weights of the commonality evaluation indexes of light and small multi-rotor UAVs.

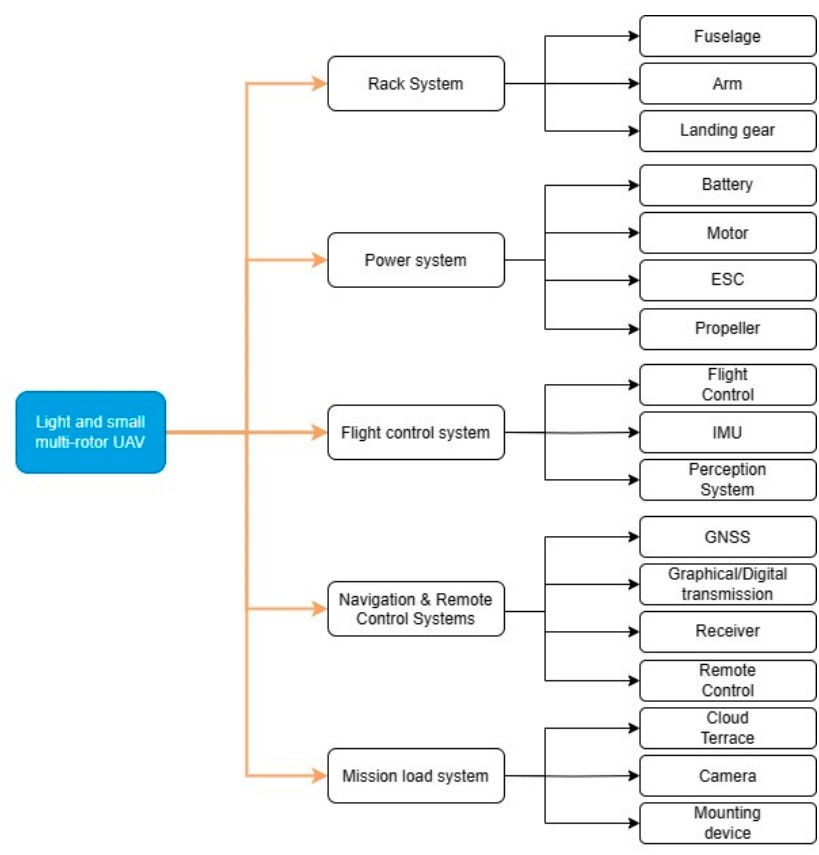

IMU:Intertidal Measurement Unit　GNSS: Global Navigation Satellite Systems　ESC:Electronic Speed Controller

**Figure 4.** Product breakdown of light and small multi-rotor UAV.

### 3.2. UAV Design Structure Matrix

Design structure matrix (DSM) refers to a network modeling tool initially proposed by Dr. Steward. This American scholar represented the elements that make up a system

and its interactions. It is used to plan and analyze the product development process, thus highlighting the design structure, and is widely used in engineering management [30]. The DSM is represented as an N × N square matrix that maps the interactions between a collection of N system elements. In modeling the architecture of a product, the DSM elements can be the components of the product. The interactions are the connections between the components, and the matrix thus composed is termed the product design structure matrix.

The modeling steps for building a matrix model of the product design structure are as follows.

(1) Determine the elements of the ranks of the DSM model in the product structure.

By decomposing the product structure, the smallest unit of the product that needs to be designed is obtained. Then the appropriate unit is selected as the element of DSM [31]. For instance, some important structural parts can be meticulously divided into the part level, while the purchased parts in the product only should be divided into the component or part levels. The selection of matrix elements in this step determines the complexity of the DSM model.

(2) Determine the connection and strength between each row and column element to obtain the DSM model of digital products.

The connection between the elements and the strength of the connection should be determined after obtaining the row and column elements of the product design structure matrix. The links between the elements of the ranks and columns in the product design structure matrix can be classified into four categories (i.e., spatial links, energy links, information links, and material links [32]), as presented in Figure 5. Spatial connection indicates the relationship between the physical space and arrangement of two elements, which describes the connection and positioning between two elements. Energy connection reveals the energy exchanged and transmitted between elements. Information connection indicates the data and signals exchanged or transmitted between two elements. Material connection indicates the material required for the exchange between two elements (e.g., the oil or gas required for the product).

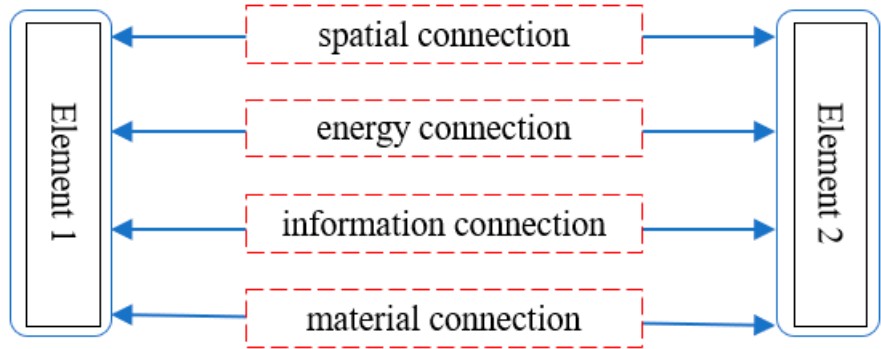

**Figure 5.** Interconnection of elements in the product Design Structure Matrix.

The four connection strengths between elements are digitized using the four-point scale method, as listed in Table 1, to obtain the product digital DSM model. Each cell in the digital DSM represents the integrated connection strength between the elements, which can be calculated using the following formula.

$$T_{i,j} = \alpha S_{i,j} + \beta E_{i,j} + \gamma I_{i,j} + \alpha M_{i,j} \tag{7}$$

where: $T_{i,j}$ is the comprehensive connection strength of the DSM cell $(i, j)$; $S_{i,j}$ is the spatial connection strength of the DSM cell $(i, j)$; $E_{i,j}$ is the energy connection strength of the DSM cell $(i, j)$; $I_{i,j}$ is the information connection strength of the DSM cell $(i, j)$; $M_{i,j}$ is the material connection strength of the DSM cell $(i, j)$; $\alpha$, $\beta$, $\gamma$, and $\omega$ indicate the relative importance of spatial connection, energy connection, information connection, and material connection,

respectively, and the appropriate values are selected according to the actual situation of the product. Finally, $T_{i,j}$ is filled into the DSM to obtain the digital product DSM model.

**Table 1.** Four-point scale method characterization and meaning.

| Graduations | Representation | Meaning |
|:---:|:---:|:---:|
| 3 | High | High connection strength |
| 2 | Medium | Medium connection strength |
| 1 | Low | Low connection strength |
| 0 | None | No connection |

(3) Check the DSM model elements and their connections, add missing connections, eliminate unnecessary connections, and complete the finalized product design structure matrix.

Because the spatial correlation matrix is symmetrical, leading to double-counting in correlation coefficients, $\alpha$ is set to 0.5. Drones using electrical energy without internal material transfer have no material contact, so $\omega$ is 0. Energy and information contact intensities are unaffected by other factors; thus, $\beta$ and $\gamma$ are set to 1. The values of each cell in the UAV design structure matrix are calculated according to Equation (7) to constitute the final UAV design structure matrix, as shown in Figure 6. The values on the diagonal of this UAV design structure matrix are the sum of all cell data in the corresponding row and column, representing the connection strength of the UAV components. For example, the connection strength of the fuselage in the frame system is 35, which is greater than that of the arm and landing gear, indicating that the fuselage occupies a more important position in the design process of the frame system and requires more consideration. Therefore, the importance of the components in the system can be expressed by calculating the contribution of each component to the system. For example, the total connection strength of the three components of the rack system is 57, and the connection strength of the fuselage is 35, so the contribution of the fuselage to the rack system is 35/57. Accordingly, the commonality evaluation indexes of light and small multi-rotor UAVs can be weighted according to the component importance defined in the UAV design structure matrix.

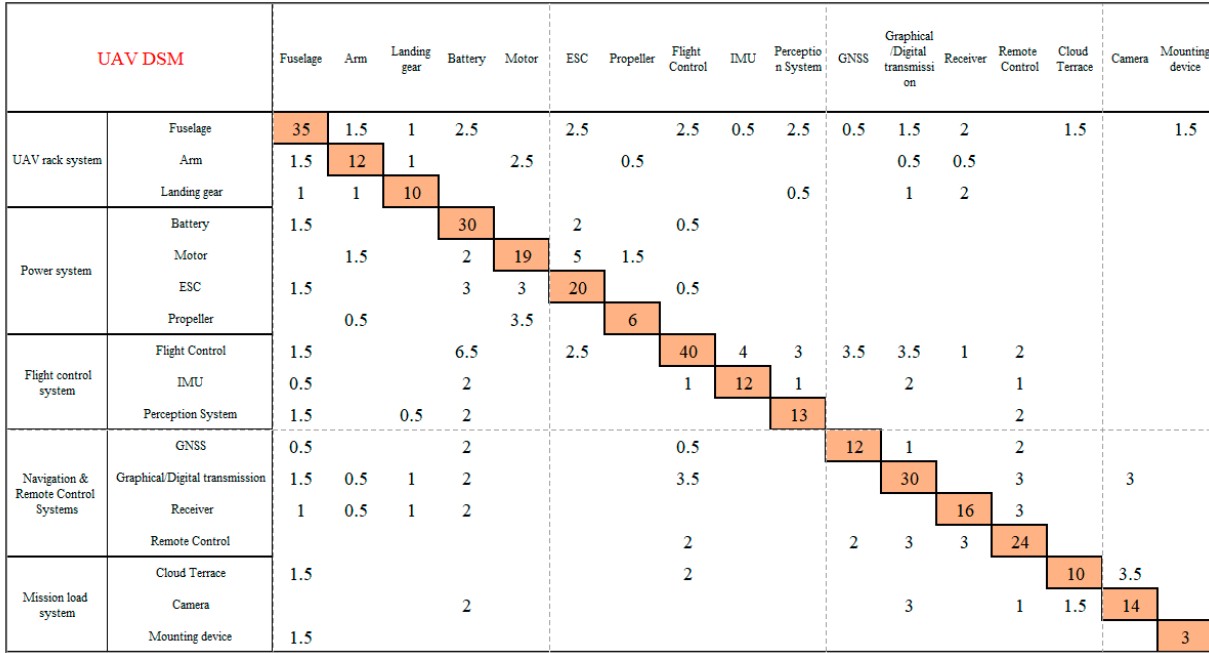

**Figure 6.** Light and small multi-rotor UAV design structure matrix. Note: The value at the off-diagonal position of the matrix indicates the correlation strength of the two items; the value at the diagonal position indicates the importance of the item.

*3.3. UAV Commonality Evaluation Indexes*

Suitable commonality evaluation indexes should be built to evaluate the commonality of UAVs. The relevant commonality evaluation indexes in current civil aircraft commonality studies [33] are referenced based on the insights gained into light and small multi-rotor UAVs and the combination of the valid UAV data that can be collected. The UAV commonality evaluation indexes presented in Figure 7 and Tables 2 and 3 are built and are primarily classified into two major parts (performance parameter commonality and structural system parameter commonality). The performance parameter commonality indexes were divided into three layers, with the second layer as the UAV performance indexes and the last layer as the evaluation feature variables to express the previous layer indexes with actual values. The structural system parameter commonality indexes are divided into four layers. To be specific, the second layer is the UAV system commonality index, the third layer is the UAV component commonality index, and the last layer is the evaluation feature variable as well. Since the parameter information of some UAV components is difficult to obtain, the performance characteristics regarding the components serve as the characteristic variables for the commonality evaluation. For instance, the maximum ascent speed, maximum descent speed, maximum horizontal flight speed, and maximum takeoff altitude serve as the characteristic variables for the motor and ESC in the commonality of the power system. Furthermore, several performance characteristics serve as the characteristic variables for the commonality evaluation of the flight control system.

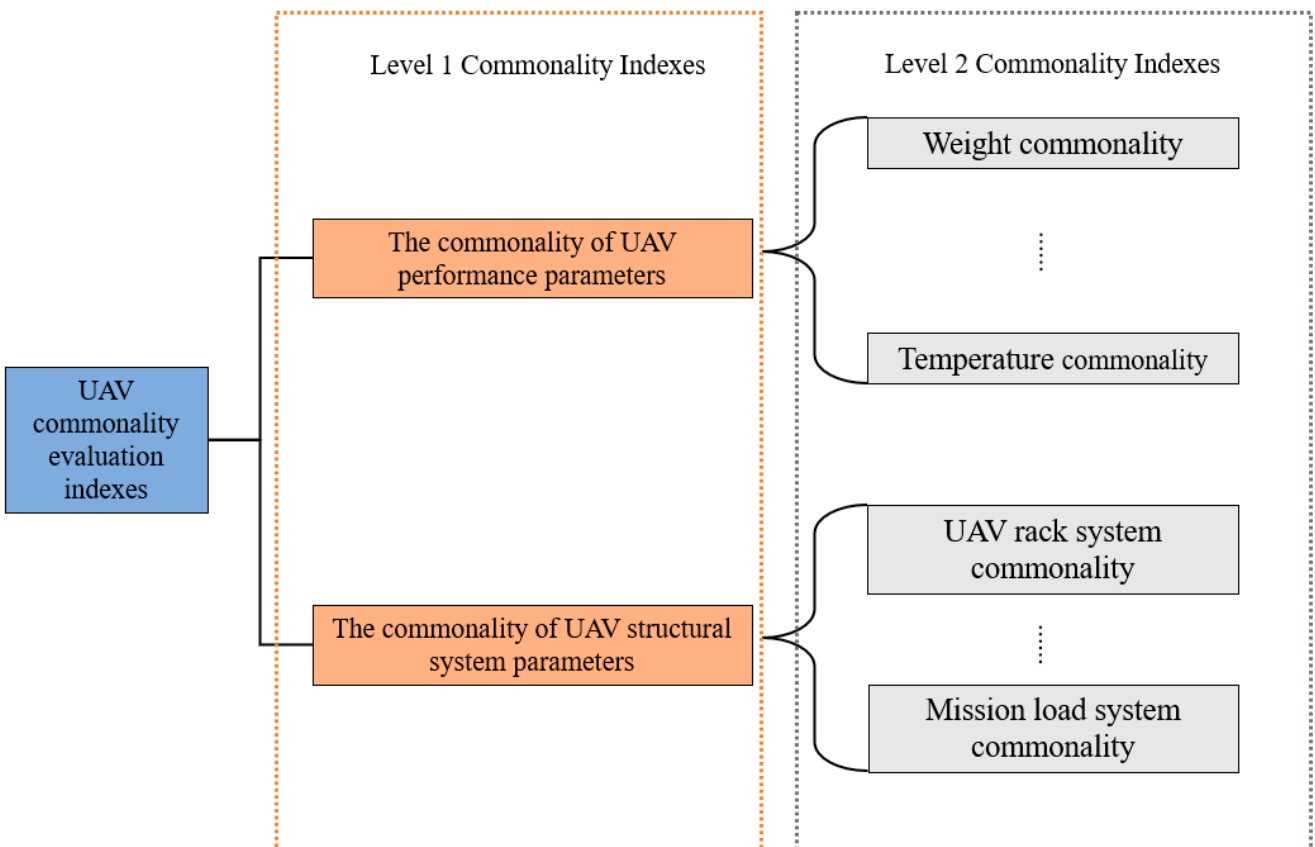

**Figure 7.** Light and small multi-rotor UAV commonality evaluation indexes.

**Table 2.** UAV performance parameters commonality index.

| Level 1 Indexes | Level 2 Indexes | Evaluation of Characteristic Variables |
|---|---|---|
| The commonality of UAV performance parameters | Weight | Body weight (g) |
| | | Maximum load weight (g) |
| | Speed — Ascent speed | Maximum ascent speed in sports gear (m/s) |
| | | Maximum ascent speed in normal gear (m/s) |
| | | Maximum ascent speed in smooth gear (m/s) |
| | Descent rate | Maximum descent speed in sports gear (m/s) |
| | | Maximum descent speed in normal gear (m/s) |
| | | Maximum descent speed in smooth gear (m/s) |
| | Horizontal flight speed | Maximum horizontal flight speed in sports gear (m/s) |
| | | Maximum horizontal flight speed in normal gear (m/s) |
| | | Maximum horizontal flight speed in smooth gear (m/s) |
| | Tilt angle | Maximum tilt angle of sports gear (°) |
| | | Maximum tilt angle of normal gear (°) |
| | | Maximum tilt angle of smooth gear (°) |
| | Flight time | Maximum endurance (min) |
| | Maximum take-off altitude | Altitude (km) |
| | Wind resistance | Wind resistance class |
| | Temperature | Minimum temperature (°C) |
| | | Maximum temperature (°C) |

**Table 3.** UAV structural system parameters commonality index.

| Level 1 Indexes | Level 2 Indexes | Level 3 Indexes | Evaluation of Characteristic Variables | |
|---|---|---|---|---|
| The commonality of UAV structural system parameters | UAV rack system commonality | Fuselage | Length of fuselage unfolding (mm) | |
| | | | Width of fuselage unfolding (mm) | |
| | | | Height of fuselage unfolding (mm) | |
| | | | Wheelbase (mm) | |
| | | | Length of fuselage folding (mm) | |
| | | | Width of fuselage folding (mm) | |
| | | | Height of fuselage folding (mm) | |
| | | UAV arm and landing gear | Number of arms | |
| | | | Arm Mounting | Foldable |
| | | | | Non-foldable |
| | | | Landing gear layout | Bottom support layout |
| | | | | Connection arm layout |
| | Power system commonality | Battery | Battery Capacity (mAh) | |
| | | | Voltage (V) | |
| | | | Energy (Wh) | |
| | | | Weight (g) | |
| | | | Charging power (W) | |
| | | Motors and ESCs | Maximum ascent speed (m/s) | |
| | | | Maximum descent speed (m/s) | |
| | | | Maximum horizontal flight speed (m/s) | |
| | | | Maximum take-off altitude (km) | |
| | | Propeller | Number of propeller blades | |
| | | | Total number of propellers | |
| | | | Propeller mounting position | Upward |
| | | | | Downward |

**Table 3.** *Cont.*

| Level 1 Indexes | Level 2 Indexes | Level 3 Indexes | Evaluation of Characteristic Variables | |
|---|---|---|---|---|
| The commonality of UAV structural system parameters | Flight control system commonality | Flight control, IUM, perception system | Perception system arrangement | Front-end Perception<br>Rear Perception<br>Lower Perception<br>Upper Perception<br>Lateral Perception |
| | | | Hovering accuracy | Vertical direction (m)<br>Horizontal direction (m) |
| | | | Maximum tilt angle (°)<br>Maximum wind resistance class | |
| | Navigation and remote control systems commonality | GNSS | GPS<br>GLONASS<br>Galileo<br>BeiDou | |
| | Navigation and remote control systems commonality | Graphical/Digital transmission/Receiver | Operating frequency | 2.4 GHz<br>5 GHz |
| | | | Data interface type | Lightning<br>Micro USB<br>Type-C<br>HDMI |
| | | | Signal effective distance (km) | FCC Distance<br>CE Distance<br>MIC Distance<br>SRRC distance |
| | | | Maximum bit rate (Mbps)<br>Delay (ms) | |
| | | Remote control | Battery capacity (mAh)<br>Operating current (mA)<br>Operating voltage (V) | |
| | Mission load system commonality | Cloud terrace | Stabilization system (number of axes)<br>Maximum control speed (°/s)<br>Amount of angular jitter (°) | |
| | | | Head structure design range (°) | Pitch angle<br>Rolling angle<br>Yaw angle |
| | | | Controllable rotation range (°) | Pitch angle<br>Rolling angle<br>Yaw angle |
| | | Camera | Pixel size (million)<br>Maximum video bit rate (Mb/s)<br>Lens angle of view (°)<br>Lens focal length (mm)<br>Lens aperture (f/X) | |
| | | | Maximum photo size | Long (PX)<br>Width (PX) |
| | | | Video resolution | HD<br>FHD<br>2.7 K<br>4 K<br>Larger than 4 K |
| | | Mounting device | Presence of mountings<br>No mountings | |

### 3.4. UAV Commonality Calculation

After the data are acquired in accordance with the UAV commonality evaluation index, the evaluation of characteristic variables data at the lowest level of the commonality evaluation index is converted into a vector and then quantified using Equation (4). Equation (6) is employed for quantification if the feature variables corresponding to this level of the index have 0–1-encoded data types. The two quantified data are averaged to determine the commonality value of the index's lowest level. With the camera under the UAV mission load system as an example, the feature data are used, as listed in Table 4, and the commonality of the camera is obtained as 0.874 after calculation.

**Table 4.** UAV Camera Commonality Evaluation Parameter Table.

| Evaluation of Characteristic Variables | Model 1 | Model 2 | Formula 4 Calculation | Formula 6 Calculation | Commonality |
|---|---|---|---|---|---|
| Pixel size (million) | 2000 | 2000 | | | |
| Maximum video bit rate (Mb/s) | 120 | 100 | | | |
| Lens angle of view (°) | 82 | 77 | | | |
| Lens focal length (mm) | 28.6 | 28 | 0.998 | \ | |
| Lens aperture (f/X) | 11 | 11 | | | |
| Long | 5472 | 5472 | | | |
| wide | 3648 | 3648 | | | 0.874 |
| HD | 0 | 0 | | | |
| FHD | 1 | 1 | | | |
| 2.7 K | 1 | 1 | \ | 0.750 | |
| 4 K | 1 | 1 | | | |
| Larger than 4 K | 1 | 0 | | | |

The commonality of the lowest level indicator is obtained based on the calculated commonality for the characteristic variable data, and the final commonality value of the UAV is quantified through weighted summation layer by layer. Subsequently, the weights of the respective indicator in the UAV commonality evaluation index should be determined. The performance—component correlation matrix is adopted to determine the UAV performance parameter commonality index weights, as listed in Table 5. To be specific, the value 3 suggests that the component significantly affects the performance, the value 2 indicates that the component exerts an average influence on the performance, the value 1 indicates that the component slightly affects the performance, and the space reveals that the component does not affect the performance. For the UAV structural system parameter commonality index weights, the UAV design structure matrix is employed to determine the weights of the third level index and the second level index using the contribution of different components to the system which they belong to and that of the respective system to the overall UAV, respectively. The final obtained indicator weights at the respective level are listed in Table 6, with the first-level indicator weights derived from the values in reference [33]. This reference, focusing on the commonality evaluation of civil aircraft, assigns a weight of 0.4 to the performance parameter commonality index and 0.6 to the structural parameter commonality index based on the relationship between performance and structure in civil aircraft. Believing that UAVs have a similar relationship, we adopted identical weights for both performance and structural parameters. Furthermore, the commonality indexes of the respective level are weighted and summed one by one to evaluate the commonality of the light and small multi-rotor UAVs. Consequently, the final degree of commonality between two models can be expressed as a percentage between 0 and 1, with a value closer to 1 indicating a higher degree of commonality between the two models.

**Table 5.** UAV performance–component correlation matrix and performance index weights.

|  | Weight | Speed | Tilt Angle | Endurance Time | Take-Off Altitude | Wind Resistance | Operating Temperature |
|---|---|---|---|---|---|---|---|
| Fuselage | 3 |  |  | 1 |  | 1 |  |
| Arm | 2 |  | 1 |  |  | 1 |  |
| Landing gear | 2 |  |  |  |  |  |  |
| Battery | 3 | 1 |  | 3 | 2 |  | 3 |
| Motor | 2 | 3 |  |  | 2 |  |  |
| ESC | 1 | 2 |  |  |  |  | 1 |
| Propeller | 1 | 3 | 2 |  | 2 |  |  |
| Flight Control | 1 | 2 | 3 |  |  | 3 |  |
| IMU | 1 |  | 3 |  |  | 2 |  |
| Perception System | 1 |  |  |  |  |  |  |
| GNSS | 1 |  |  |  |  |  |  |
| Graphical/Digital transmission | 1 |  |  |  |  |  |  |
| Receiver | 1 |  |  |  |  |  |  |
| Remote Control |  |  |  |  |  |  |  |
| Cloud Terrace | 1 |  |  | 1 |  |  |  |
| Camera | 1 |  |  | 1 |  |  |  |
| Σ | 23 | 11 | 9 | 7 | 6 | 7 | 4 |
| Weights | 0.343 | 0.164 | 0.134 | 0.105 | 0.09 | 0.104 | 0.06 |

**Table 6.** Table of the weighting of indexes at the respective level in the UAV commonality evaluation system.

| Performance Commonality (0.4) | Structural System Commonality (0.6) | | | | |
|---|---|---|---|---|---|
| Weight (0.343)<br>Speed (0.164)<br>Tilt angle (0.134)<br>Endurance time (0.105)<br>Take-off altitude (0.09)<br>Wind resistance (0.104)<br>Operating temperature (0.06) | UAV rack system (0.186) | Power system (0.245) | Flight control system (0.212) | Navigation and remote control system (0.268) | Mission load system (0.088) |
| | Fuselage (0.614)<br>Arm and landing gear (0.386) | Battery (0.400)<br>Motor and ESC (0.520)<br>Propeller (0.080) | Flight control, IUM, Perception system (1) | GNSS (0.146)<br>Graphical/Digital transmission/Receiver (0.561)<br>Remote control (0.293) | Cloud terrace (0.370)<br>Camera (0.519)<br>Mounted devices (0.013) |

*3.5. Case Calculation*

DJI and Autel refer to two large Chinese manufacturers of light and small multi-rotor UAVs, both of which have developed UAV product series; their products show high similarity. Thus, in this study, three models of DJI's UAVs (Mavic 2, Mavic 3, and Phantom 4 Pro) and one model of Autel's UAV (Autel EVO ll Pro) serve as the computational case of UAV commonality evaluation, and the four models are illustrated in Figure 8.

The evaluation characteristics data of all four types of models are collected based on the UAV commonality evaluation index. Six groups of samples of the four types of models are set in two groups as the commonality evaluation calculation cases. The commonality values of UAV, UAV performance parameters, UAV structural system parameters, and five subsystems of UAV are obtained after calculation using the UAV commonality evaluation model. The specific data are illustrated in Figure 9.

As depicted in Figure 9, the two models of the DJI Mavic series exhibit the highest commonality value of 0.873, notably higher than the other five groups. Since Mavic 3 refers to the latest product currently developed by DJI based on Mavic 2, enhancing the flight performance of the UAV, the overall architecture of the airframe remains unchanged and focuses on performance upgrades of the gimbal, camera, and other components of the UAV mission payload system. On that basis, the frame system exhibits a high commonality in

the commonality evaluation results of Mavic 2 and Mavic 3, whereas the lowest mission payload commonality is achieved, consistent with the actual situation of the model.

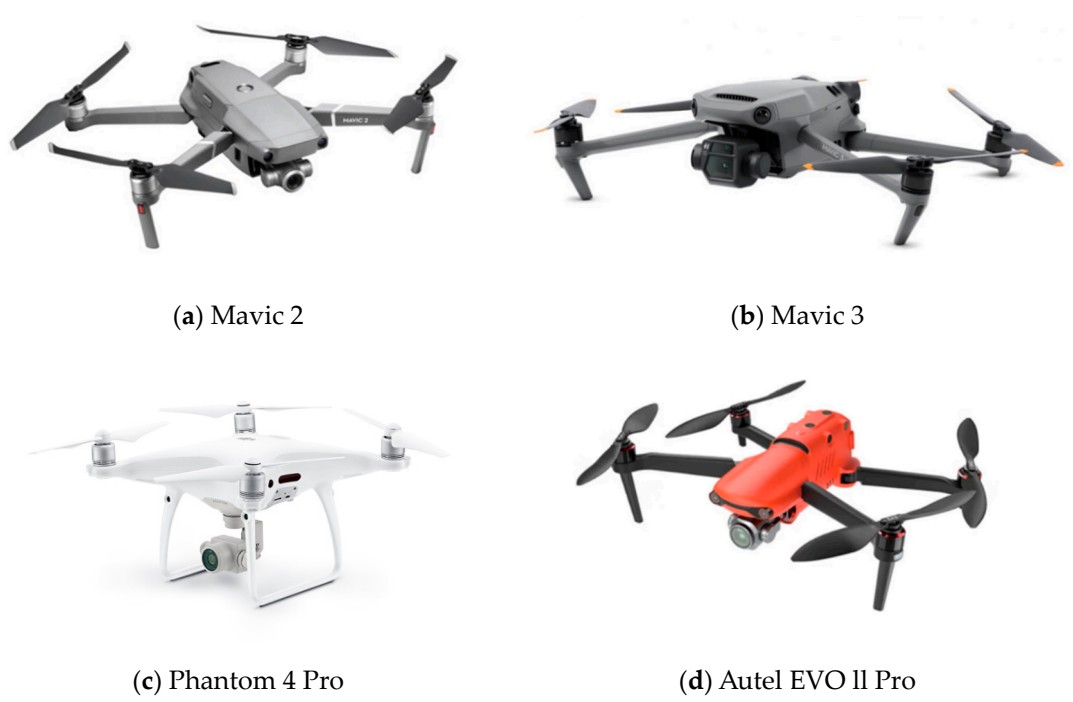

(**a**) Mavic 2

(**b**) Mavic 3

(**c**) Phantom 4 Pro

(**d**) Autel EVO ll Pro

**Figure 8.** Four types of light and small multi-rotor UAV models.

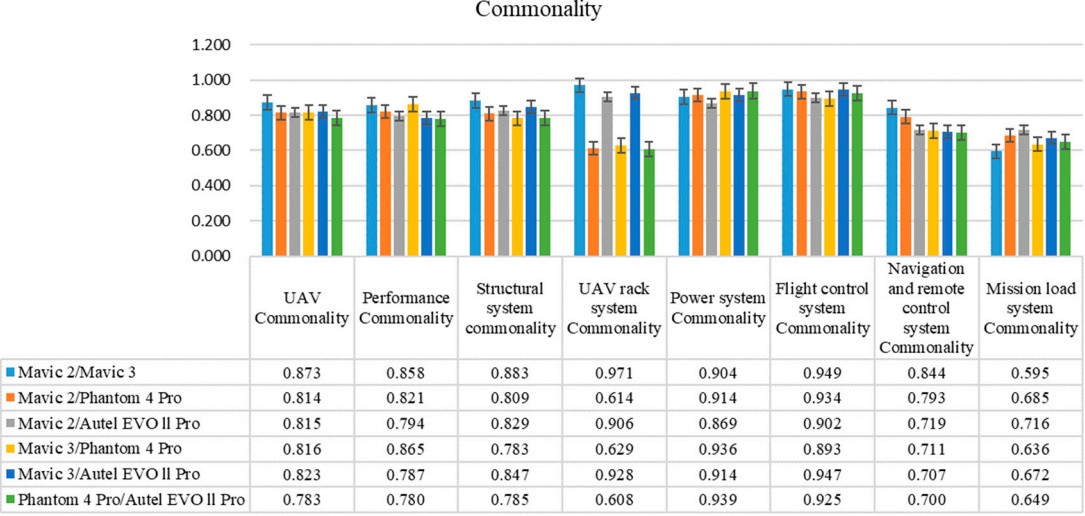

| | UAV Commonality | Performance Commonality | Structural system commonality | UAV rack system Commonality | Power system Commonality | Flight control system Commonality | Navigation and remote control system Commonality | Mission load system Commonality |
|---|---|---|---|---|---|---|---|---|
| Mavic 2/Mavic 3 | 0.873 | 0.858 | 0.883 | 0.971 | 0.904 | 0.949 | 0.844 | 0.595 |
| Mavic 2/Phantom 4 Pro | 0.814 | 0.821 | 0.809 | 0.614 | 0.914 | 0.934 | 0.793 | 0.685 |
| Mavic 2/Autel EVO ll Pro | 0.815 | 0.794 | 0.829 | 0.906 | 0.869 | 0.902 | 0.719 | 0.716 |
| Mavic 3/Phantom 4 Pro | 0.816 | 0.865 | 0.783 | 0.629 | 0.936 | 0.893 | 0.711 | 0.636 |
| Mavic 3/Autel EVO ll Pro | 0.823 | 0.787 | 0.847 | 0.928 | 0.914 | 0.947 | 0.707 | 0.672 |
| Phantom 4 Pro/Autel EVO ll Pro | 0.783 | 0.780 | 0.785 | 0.608 | 0.939 | 0.925 | 0.700 | 0.649 |

**Figure 9.** Commonality evaluation results.

Comparing the Autel EVO ll Pro UAV with DJI's Mavic series UAVs in Figure 8, we can see that they have high similarity in appearance. If subjective commonality evaluation is used, people usually judge that they have a high degree of commonality with each other. The commonality evaluation model of the light and small multi-rotor UAVs has calculated the commonality values of 0.815 and 0.823 between the Autel EVO ll Pro and the Mavic 2 and Mavic 3, respectively, which are indeed high commonality. Because both Autel EVO ll Pro and Mavic 3 are novel products in the current market, the product development time is similar and the UAV performance is more similar, so it causes their commonality to be slightly higher.

Comparing the appearance of the Phantom 4 Pro UAV with the other three models in Figure 8, it is found that the overall shape and general layout of the Phantom 4 Pro

UAV differs significantly compared to the other three models. The commonality evaluation model calculates that the Phantom 4 Pro UAV has a low commonality value with the rack systems of the other models. Meanwhile, the lowest commonality value between the Phantom 4 Pro and the Autel EVO ll Pro was calculated using the model, which is due to the fact that the two above-mentioned models are not only different in overall layout but also developed by different manufacturers, resulting in the weakest commonality, which is in line with the basic theory of commonality and basic public perception.

In summary, the commonality calculated using the light and small multi-rotor UAV commonality evaluation model is consistent with UAVs' actual comparison results, indicating that the built light and small multi-rotor UAV commonality evaluation model can be better applied to the commonality evaluation of light and small multi-rotor UAVs.

## 4. The Prediction Model Based on Convolutional Neural Networks for the Commonality of Light and Small Multi-Rotor UAVs

### 4.1. Data Collection and Cleaning

To obtain the parameters of the light and small multi-rotor UAVs listed in Tables 2 and 3, this study collects model characteristics data from the official websites or official online stores of domestic and foreign UAV manufacturers based on the commonality evaluation index of light and small multi-rotor UAVs. A total of 46 models of UAVs were identified through web search, including UAV models of DJI, Parrot, and other companies. Because a large amount of feature data could not be obtained for some models, 24 typical light and small multi-rotor UAV models were finally selected from the 46 models. The distribution of the specific sources is listed in Table 7.

**Table 7.** Source distribution of models.

| UAV Brands | DJI | Autel | Hubsan | Parrot |
| --- | --- | --- | --- | --- |
| Number of models | 16 | 3 | 3 | 2 |

According to the commonality evaluation index of light and small multi-rotor UAVs, the characteristic data of the above 24 models are relatively large, but not all characteristic data of each model can be obtained completely. Since the data are all collected manually, the directly obtained data are accurate and reliable. Thus, no noise data are generated and no operations such as data rejection are required, but there will be a large amount of vacant data. Accordingly, completing the missing data becomes the top priority of data cleaning, and the following is the lost-data-completing method adopted in this study.

(1) Reasoned deduction method: When certain data cannot be directly obtained, we will reasonably infer based on the information provided in the UAV's user manual and images. For example, when filling in data for a UAV's ascent speed, descent speed, and cruising speed, if a model does not distinguish between flight modes, the corresponding speed data will be filled under the primary flight mode, and data for other modes will be supplemented as zero. If a UAV model lacks sensor location parameters in its perception system, this data can be deduced by analyzing the position of sensors in UAV images.

(2) Analogy supplement method: When certain parameters cannot be derived from the user manual or images of the UAV model, we can refer to the corresponding parameters of the same series from the same manufacturer. For example, in cases where remote control feature data of a UAV are missing, data from the remote controls of other UAVs of the same series by the same company can be used for supplementation.

(3) Mode and mean imputation method: When neither of the above methods is applicable, mode imputation is used, and mean imputation is employed when the mode is not a unique value. For instance, in cases in which data on a UAV's maximum takeoff altitude or wind resistance level are missing, the mode is commonly used as a substitute value, such as supplementing the maximum takeoff altitude as 5 km and the wind resistance level as level five.

### 4.2. Constructing the Dataset

After processing, the 24 types of model characteristics data are compared two by two to form a set of samples, and a total of 276 samples are obtained. Subsequently, the sample data are substituted into the commonality evaluation model of light and small multi-rotor UAVs to obtain the commonality value of the respective group of samples. The number of commonality interval distributions of 276 groups of samples is presented in Figure 10, and the distribution conforms to the normal distribution curve corresponding to Equation (8) (orange curve in Figure 10).

$$f(x) = \frac{1}{\sqrt{2\pi} \times \sqrt{0.061}} \exp\left(-\frac{(x - 0.779)^2}{2 \times 0.061}\right) \tag{8}$$

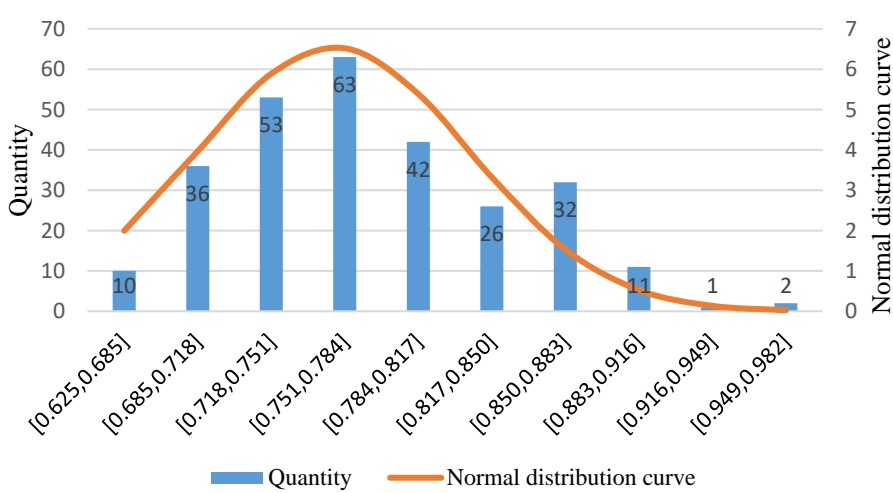

**Figure 10.** Distribution of commonality values for the sample of light and small multi-rotor UAV commonality prediction.

According to the UAS analysis process shown in Figure 11, when developing a novel model UAV, it is necessary first to conduct a requirement analysis, which is generally divided into two sources as follows. One is the customer's customized requirements, and the other is to determine the requirements through the market demand analysis; then, the UAV product performance analysis is conducted in accordance with the requirements, and the user's requirements are converted into the performance indexes of the UAV. Then, through the architecture analysis, the UAV system is determined. Next, the organizational structure of the UAV system and the logical relationships between the systems are determined through architecture analysis. Lastly, the UAV's hardware is determined and the UAV's physical architecture model is built. The above-described process requires repeated iterations of verification and eventually forms a UAV product that conforms to the requirements.

To achieve the purpose of predicting the commonality through a small amount of feature data at the early stage of UAV design, suitable features from the commonality evaluation index should be selected as feature variables for the commonality prediction dataset of the light and small multi-rotor UAV. From the above system analysis process analysis, it can be seen that the UAV design is based on requirements and performance as indexes; combined with the UAV life cycle, it can be seen that the general design parameters and performance parameters of the UAV can be determined when the UAV completes the program design. Thus, this study selects the feature variables as listed in Table 8, which are mainly divided into two categories: one for the general design features of the UAV and the other for the performance features of the UAV, where the three features of battery capacity,

longest data transmission distance, and camera pixels in the second category are selected based on user requirements.

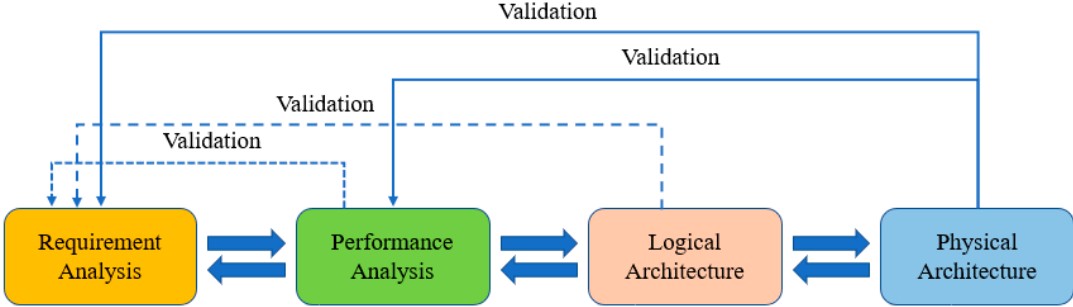

**Figure 11.** System analysis flow.

**Table 8.** Characteristics variables of the light and small multi-rotor UAV commonality prediction dataset.

| General Design Features of UAV | UAV Performance Characteristics |
|---|---|
|  | Maximum ascent speed |
|  | Maximum descent speed |
| Body weight | Maximum horizontal flight speed |
| Number of arms | Maximum tilt angle |
| Fuselage spread length | Maximum flight time |
| Width of fuselage spread | Maximum takeoff altitude |
| Fuselage unfolded high | Wind resistance class |
| Wheelbase | Battery capacity |
|  | Longest data transmission distance |
|  | Camera pixels |

Lastly, the feature amount of the selected light and small multi-rotor UAV commonality prediction dataset in Table 8 takes up 17% of the total feature amount of the light and small multi-rotor UAV commonality evaluation index, completing the selection of a small number of features for light and small multi-rotor UAVs. Subsequently, the feature variable data of the selected 24 models are formed into a group of two, and the evaluated commonality values of the respective group are added to collectively create a commonality prediction dataset of light and small multi-rotor UAVs with 276 samples. To be specific, the feature data of the two compared models serve as the independent variables, and the commonality values serve as the dependent variables. The webpage to download the dataset is https: //github.com/Amos111/Commonality-Prediction-Dataset (accessed on 18 February 2023).

*4.3. Convolutional Neural Network Model Building*

In the process of CNN model construction, if the built model is highly complex, the phenomenon of overfitting will occur. For instance, the overfitting is shown in Figure 12, where the training error of the model is minimal, whereas the generalization error is high, suggesting poor application of the model. If the built model is too simple, the phenomenon of underfitting will occur. For instance, the underfitting is illustrated in Figure 12, in which the training error and generalization error of the model are both high, revealing the poor application of the model poor overall quality [34]. Accordingly, a model of suitable complexity with both low generalization error and appropriate training error should be built.

For convolutional neural networks, the main objective of adjusting model complexity is fulfilled by adjusting the number of model layers and convolutional kernels [35]. After continuous experiments, the CNN structural model listed in Table 9 is finally built; the model structure includes 13 layers of networks, and the detailed parameters of each layer are given in the table. After the CNN structural model is built, the model parameters should

be set to train the data. Adjusting the model parameters is critical to increasing the model learning effect. The adopted model training parameters are determined after continuously adjusting the model parameters, as listed in Table 10. The optimization algorithm used is the Adam algorithm; the batch size for small batch training is 100 samples, the maximum instances of training times is 800, and the initial learning rate is set to 0.001.

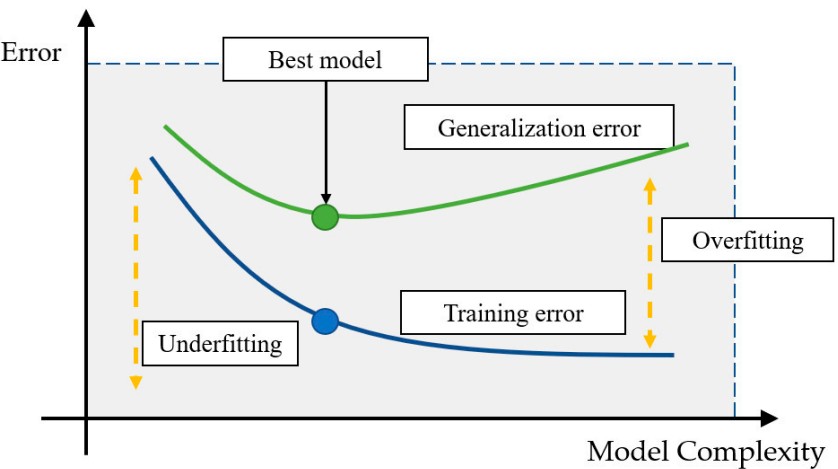

**Figure 12.** Optimal model schematic diagram.

**Table 9.** CNN structure for light and small multi-rotor UAV commonality prediction.

| Layer | Type | Parameters | Neurons | Output |
|-------|------|------------|---------|--------|
| 1 | Input layer | | | $16 \times 2 \times 1$ |
| 2 | Convolutional layer | $2 \times 2$ convolution kernel | 256 | $15 \times 1 \times 256$ |
| 3 | Batch normalization layer | | | $15 \times 1 \times 256$ |
| 4 | Relu layer | | | $15 \times 1 \times 256$ |
| 5 | Convolutional layer | $3 \times 1$ convolution kernel | 128 | $13 \times 1 \times 128$ |
| 6 | Batch normalization layer | | | $13 \times 1 \times 128$ |
| 7 | Relu layer | | | $13 \times 1 \times 128$ |
| 8 | Convolutional layer | $3 \times 1$ convolution kernel | 128 | $11 \times 1 \times 128$ |
| 9 | Batch normalization layer | | | $11 \times 1 \times 128$ |
| 10 | Relu layer | | | $11 \times 1 \times 128$ |
| 11 | Dropout layer | 0.2 | | $11 \times 1 \times 128$ |
| 12 | Fully connected layer | | | $1 \times 1 \times 1$ |
| 13 | Regression output layer | | | $1 \times 1 \times 1$ |

**Table 10.** Model training parameters settings.

| Optimization Algorithm | Adam |
|------------------------|------|
| MiniBatchSize | 100 |
| MaxEpochs | 800 |
| InitialLearnRate | 0.001 |
| LearnRateDropFactor | 0.1 |
| Shuffle | Yes |

*4.4. Experimental Results and Analysis*

The dataset is divided into a training set and a test set, with 80% of the total samples in the training set and the rest in the test set. The CNN model was built and programmed in Matlab language, and after training the model several times, the model with the best-combined training effect and model generalization effect was saved; the comparison of the actual and predicted values of the training and test sets of the model is shown in Figures 13 and 14.

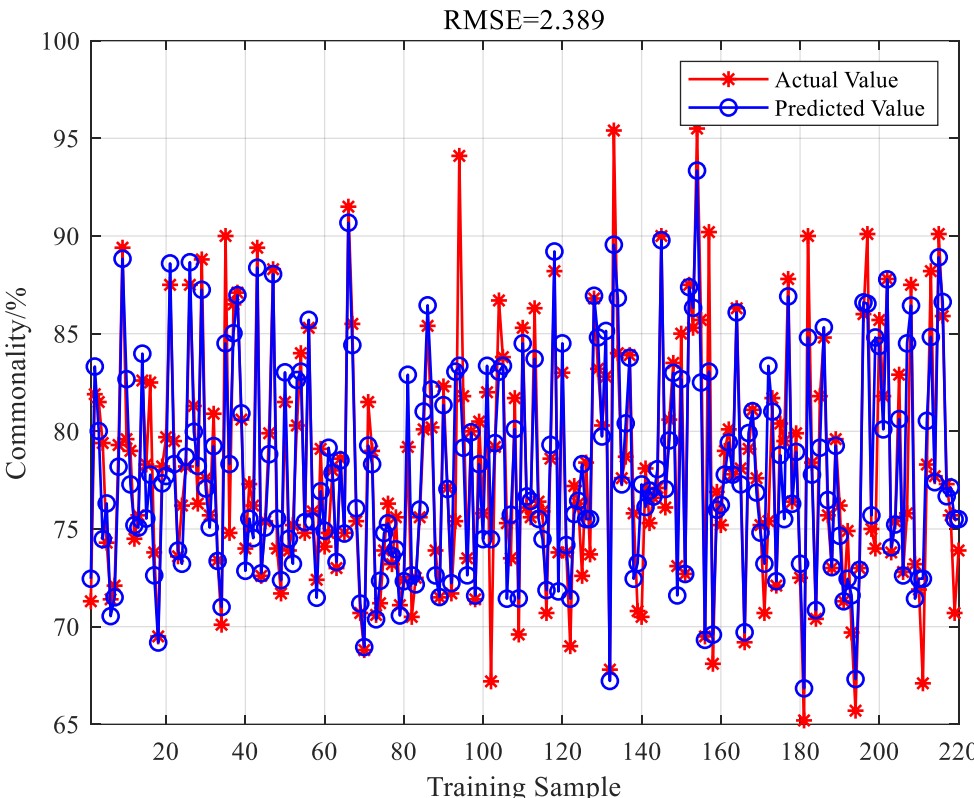

**Figure 13.** Comparison of the prediction results of the training set.

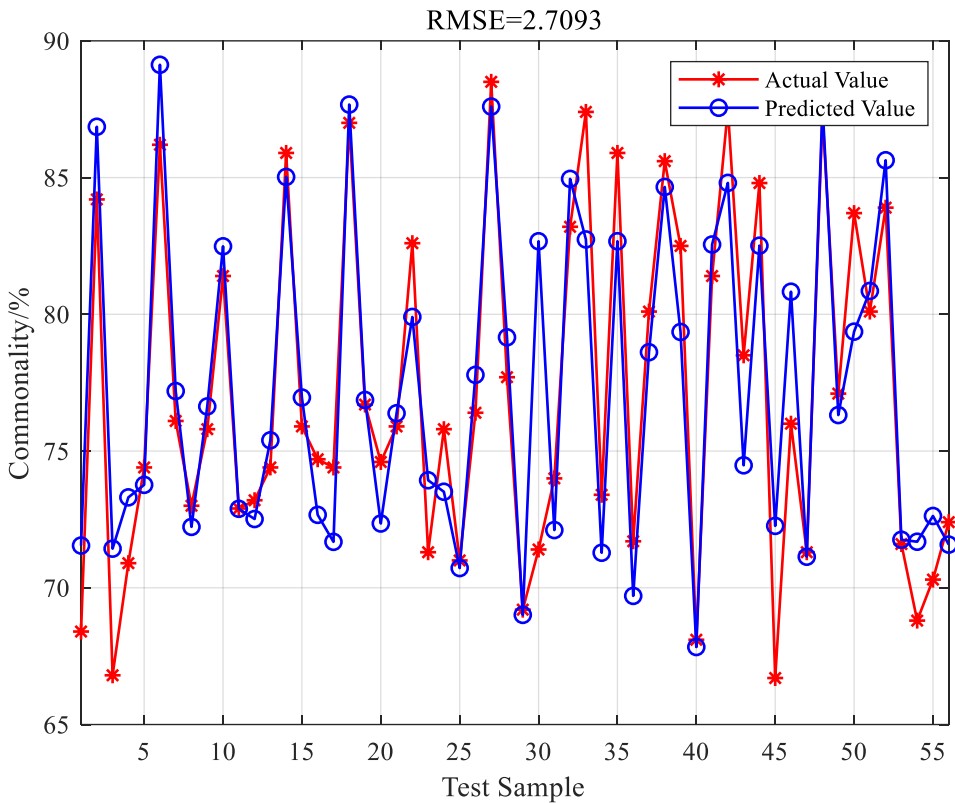

**Figure 14.** Comparison of the prediction results of the test set.

Choosing appropriate evaluation metrics is crucial in assessing model performance. Although individual evaluation metrics often have inherent limitations, they can provide

a more comprehensive perspective on performance when used together. In light of this, our study employs three different dimensional evaluation metrics to comprehensively measure the model's performance. These metrics include mean relative error (MRE), root mean square error (RMSE), and the coefficient of determination ($R^2$), which evaluate the model from the perspectives of relative error, absolute error, and model fitting effectiveness, respectively.

Mean relative error (MRE) assesses the model's prediction accuracy by calculating the proportional difference between predicted and actual values. A key advantage of this metric is its scale independence, meaning that MRE can provide consistent evaluation results regardless of the magnitude of the data. The formula to calculate MRE is:

$$MRE = \frac{1}{n}\sum_{i=1}^{i=n}\left|\frac{y_i - \hat{y}_i}{y_i}\right| \tag{9}$$

Root mean square error (*RMSE*) fundamentally measures the accuracy of predictions by calculating the square root of the average squared prediction errors. The advantage of this method is that it amplifies and emphasizes larger errors, as squaring the error values increases the weight of larger errors. *RMSE* is an absolute error metric, and the closer its value is to zero, the better the performance of the predictive model. The formula for calculating *RMSE* is:

$$RMSE = \sqrt{MSE} = \sqrt{\frac{1}{n}\sum_{i=11}^{i=n}(y_i - \hat{y}_i)^2} \tag{10}$$

The coefficient of determination, $R^2$, is a metric that reflects the degree of fit of the model to the data, measuring the proportion of variation explained by the model in relation to the total variation. The value of $R^2$ ranges from 0 to 1, with a value close to 1 indicating that the model can well explain the variation in the data, while a value close to 0 suggests weak explanatory power of the model. The calculation method for $R^2$ is:

$$R^2 = 1 - \frac{\sum(y - \hat{y})^2}{\sum(y - \overline{y})^2} \tag{11}$$

The evaluation metrics of the optimal model obtained after multiple trainings are shown in Table 11. From the *MRE* results, the model's predicted values have a relative error of about 1.93% in the test set, slightly increasing to 3.22%, which is a relatively low error rate, indicating high prediction accuracy during training. The *RMSE* results show small absolute differences between predicted and actual values in both training and test sets, around 0.25, further indicating good prediction precision of the model. In terms of the coefficient of determination, the model explains about 85.35% of the data variance in the training set, which is a relatively high fit, demonstrating good performance on the training set. In the test set, the $R^2$ value is 79.81%, slightly lower than in the training set, indicating a slight decrease in the model's ability to fit new data but still showing good performance.

**Table 11.** Quantitative evaluation of model training effects.

|  | Training Set | Test Set |
|---|---|---|
| MRE | 0.0193 | 0.0322 |
| RMSE | 0.0239 | 0.0271 |
| $R^2$ | 0.8535 | 0.7981 |

Overall, these metrics indicate that the model performs well on the training set and also demonstrates considerable generalization ability on the test set. Although there is a slight decline in metrics on the test set, which is common in model training as models always try to adapt to training data, the test set provides a measure of its general-

ization capability. Overall, the model exhibits high accuracy and good fitting ability, making it suitable for predicting the commonality evaluation values of light small-sized, multi-rotor UAVs.

*4.5. Test Cases*

　　Suppose a UAV company plans to develop a new light and small multi-rotor UAV. After the design team's discussion and preliminary research on the market, it makes the expected targets for UAV-related characteristics, as listed in Table 12. The 24 models used to build the commonality prediction dataset for light and small multi-rotor UAVs in this study are all mainstream models in the UAV market and have achieved commercial success. Thus, one of the 24 models selected as the reference object or competitive target for developing a new UAV can reduce the difficulty of entering the market for the product and make it easier to achieve market access.

**Table 12.** Data for novel model features variables.

| General Design Features | Value | UAV Performance Features | Value |
|---|---|---|---|
| Maximum take-off weight (g) | 1000 | Maximum ascent speed (m/s) | 9 |
| Number of arms | 4 | Maximum descent speed (m/s) | 7 |
| Length of fuselage unfolding (mm) | 380 | Maximum horizontal flight speed (m/s) | 25 |
| Width of fuselage unfolding (mm) | 300 | Maximum tilt angle (°) | 40 |
| Height of fuselage unfolding (mm) | 120 | Maximum flight time (min) | 40 |
| Wheelbase (mm) | 400 | Maximum take-off altitude (km) | 6 |
| | | Wind resistance class | 6 |
| | | Battery capacity (Ah) | 4.5 |
| | | Maximum transmission distance (km) | 7 |
| | | Camera Pixels (millions) | 4.8 |

　　With the built prediction model for the commonality of light and small multi-rotor UAVs, the novel model's characteristic data and the characteristic data of 24 models were composed as the model's input, and the commonality evaluation results were obtained, as listed in Table 13.

**Table 13.** Commonality prediction results.

| Target Model | Benchmark Models | Commonality (%) | Target Model | Benchmark Models | Commonality (%) |
|---|---|---|---|---|---|
| Novel model | DJI Mini SE | 77.7 | Novel model | DJI Inspire 1 | 75.9 |
| | DJI Mavic air | 76.9 | | DJI M30 | 80.2 |
| | DJI Mavic 2 | 87.0 | | DJI M300 | 75.7 |
| | DJI Mini 3 Pro | 79.8 | | DJI M200 | 76.0 |
| | DJI Mavic 3 | 89.9 | | Autel EVO ll Pro | 82.4 |
| | DJI Air 2S | 84.5 | | Autel EVO ll Lite+ | 87.7 |
| | DJI Mavic Air 2 | 84.9 | | Autel EVO NANO | 79.4 |
| | DJI Mini 2 | 79.4 | | Habsen ACE pro | 80.9 |
| | DJI Avata | 71.1 | | Habsen zinomini SE | 78.5 |
| | DJI FPV | 79.0 | | Habsen zinomini pro | 78.0 |
| | DJI Phantom 4 Pro | 83.2 | | parrot ANAFI Ai | 80.1 |
| | DJI Inspire 2 | 76.9 | | parrot ANAFI-USA | 78.0 |

　　From the data in the table, we observed that the new UAV model has an average commonality evaluation value of 80.1% with the existing 24 models, placing it in the middle-to-latter part of the multi-rotor UAV commonality distribution. This indicates that the model maintains a certain level of universality in the overall UAV market. This result suggests that the design objectives of the model are relatively universal and not as likely to encounter significant technical design challenges as anticipated. Additionally, within the light multi-rotor UAV field, there are various models available for design reference.

Particularly noteworthy is that the new model has a high commonality evaluation value with the DJI Mavic and DJI Air series, which may indicate that the new model will compete with these two series in the market. Therefore, in the product design process, to highlight the uniqueness of the new model and secure a market position, it should seek to surpass the existing series not only in meeting the predetermined technical parameters but also in aspects like appearance and reliability. This innovation and optimization in design will be key to the new model's success, enabling it to stand out in the fiercely competitive market.

The application and analysis of this case demonstrate that the method proposed in this study provides UAV design teams with a quantitative tool to assess the correlation between design concepts and existing products and also offers data support for market analysis and product positioning. By comparing UAVs of different brands and models, manufacturers can more accurately position new products, designing UAVs that meet market needs while being competitive. Therefore, the commonality evaluation method proposed in this study is not only innovative in theory but also significantly strategic in practical application.

## 5. Summary and Outlook

This study focuses on the quantitative evaluation and prediction methods of commonality in light small-sized, multi-rotor UAVs, aiming to provide UAV manufacturers with a scientific and reliable set of evaluation tools. This will assist manufacturers in achieving the optimal balance between commonality design and innovative performance, thus enhancing product design and market competitiveness. From this research, we have drawn the following conclusions:

(1) The commonality quantification method developed in this study, based on distance measurement and similarity, demonstrates high flexibility and accuracy in applications involving different types of feature data, offering an innovative solution for quantifying UAV commonality.

(2) The multi-level evaluation indicators constructed, combined with the UAV product design structure matrix, effectively achieve a comprehensive evaluation of the commonality of light small-sized, multi-rotor UAVs. The practicality and effectiveness of this method have been confirmed through empirical analysis of mainstream UAV models in the market.

(3) By establishing and training a convolutional neural network model, this study has successfully predicted the commonality of light small-sized, multi-rotor UAVs based on limited feature data, marking a significant step forward in the field of commonality prediction at the early stages of UAV design.

While the commonality evaluation and prediction models for light small-sized, multi-rotor UAVs established in this paper show notable applicability, there is still room for further research in the future. For instance, integrating non-structural features such as design concepts and processes into the evaluation system can enhance the precision of evaluation results; increasing the amount of sample data and applying deep learning techniques can improve the accuracy of prediction models; further exploring the economic benefits brought by commonality design in UAVs. We believe that these future research directions will provide more comprehensive and in-depth insights into the commonality design of light small-sized multi-rotor UAVs and offer practical and efficient solutions and decision support for the industry.

**Author Contributions:** Conceptualization, Y.Z. (Yongjie Zhang) and K.C.; methodology, Y.Z. (Yongjie Zhang) and Y.Z. (Yongqi Zeng); software, Y.Z. (Yongqi Zeng); validation, K.C.; formal analysis, Y.Z. (Yongjie Zhang); investigation, Y.Z. (Yongqi Zeng); resources, Y.Z. (Yongjie Zhang); data curation, Y.Z. (Yongqi Zeng); writing—original draft preparation, Y.Z. (Yongqi Zeng); writing—review and editing, K.C.; visualization, K.C.; supervision, Y.Z. (Yongjie Zhang); project administration, Y.Z. (Yongjie Zhang). All authors have read and agreed to the published version of the manuscript.

**Funding:** This research received no external funding.

**Data Availability Statement:** https://github.com/Amos111/Commonality-Prediction-Dataset (accessed on 18 February 2023).

**Conflicts of Interest:** The authors declare no conflict of interest.

## Nomenclature

| | |
|---|---|
| $X, Y$ | data of a set of comparison samples |
| $d_{XY}$ | Minkowski distance |
| $d_{\max}$ | maximum possible Minkowski distance between $X$ and $Y$ |
| $s_{XY}$ | Cosine similarity |
| $CI$ | value of commonality evaluation |
| DSM | design structure matrix |
| $T_{i,j}$ | comprehensive connection strength of the DSM cell $(i, j)$ |
| $S_{i,j}$ | spatial connection strength of the DSM cell $(i, j)$ |
| $E_{i,j}$ | energy connection strength of the DSM cell $(i, j)$ |
| $I_{i,j}$ | information connection strength of the DSM cell $(i, j)$ |
| $M_{i,j}$ | material connection strength of the DSM cell $(i, j)$ |
| CNN | convolutional neural network |
| $MRE$ | mean relative error |
| $RMSE$ | root mean square error |
| $R^2$ | goodness of fit |

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
