# Peer review of "Commonality Evaluation and Prediction Study of Light and Small Multi-Rotor UAVs"

_drones, doi:10.3390/drones7120698_

Round 1
Reviewer 1 Report
Comments and Suggestions for Authors
The authors propose a generalizability assessment and prediction methodology for UAVs and construct its model to predict generalizability in the early stages of multi-rotor flightless vehicle design. Before the paper is published, I suggest the following revisions:
1. the authors used a convolutional neural network to predict the generalizability of UAVs. How was the accuracy of the model ensured?
2. the authors have made side-by-side comparisons of different aerial vehicles and given comparisons of the predicted results, but some of the results lack proper explanatory notes. The authors should revise the presentation of the results appropriately and elaborate on the deeper meaning of the results.
3. the distribution of the true and predicted values of the training and test sets are shown in Figures 12 and 13, and their RMSE metrics are listed. The authors should explain how good or bad the results are or how the metrics compare to other methods, etc.
4. the authors did a lot of tests to show the prediction results of UAVs and lacked comparison results with current methods.
5. the format of the article should be modified to improve readability: e.g., figure captions for Figures 5 and 6; Table 3 should be scaled down to one page if possible; the orientation of the text in Figure 3 should be adjusted appropriately for ease of reading; the size of equations, etc.
6. The manuscript conducts a review of related work, but most of the references cited are ten years old. The authors should search for the availability of relevant literature from recent years and cite it appropriately.
Author Response
Dear Reviewer:
I am very grateful for your comments and valuable feedback, which has helped us improve some of the narratives in the article and greatly enhance the quality of the manuscript. The responses to all questions and the revised manuscript are detailed in the attachment. If you have any other opinions or questions, please contact us and we will reply as soon as possible.
Yours sincerely,
Kang Cao and Yongjie Zhang
2023.11.29

Reviewer 2 Report
Comments and Suggestions for Authors
The article proposes some regular approach to choosing the characteristics of a newly designed UAV. The idea is to write out the main characteristics in the form of a vector and then compare the corresponding vectors to determine the similarity measure.
In my opinion, this approach is not without interest, although it is rather controversial, since the classification according to the area of purpose of the device is more obvious. If the authors could connect their classification method with the implementation of the most interesting UAVs, such an article might be of more interest to the reader of the journal.
Comments on the Quality of English Language
More or less adequate.
Author Response

(The authors gave the same response as above.)

Reviewer 3 Report
Comments and Suggestions for Authors
This paper uses the Design Structure Matrix theory and composition architecture elements to build the commonality evaluation model and index for lightweight and small multirotor UAVs. The authors have used the data of typically light and small multi-rotor UAV models to build the commonality prediction dataset of light and small multi-rotor UAVs, which is used to predict the commonality at the early stage of multi-rotor UAV design with new light and tiny multi-rotor UAV models. As a result, the work is worthy of publication if the remarks listed are taken into account:
1- First of all, the abstract does not reflect the importance of the study properly, so I think it has to be strengthened to receive more concern from the academic world.
2- Although you have utilized the Commonality quantification method, it is unclear what the common domain values are. Could you kindly clarify the values indicated?
3- Unfortunately, I'm not sure what is new in light of the weak abstract and lack of a conclusion (the summary does not imply a conclusion). Thus, more justification is required along with a helpful conclusion.
4- Why didn't you use the Endurance Performance Rapid Evaluation Model and Apply It to UAVs? Is there any reason? Considering what you said in your comment, you can also include a brief explanation of why you chose an alternative method to the Rapid Evaluation Model.
5- Even if the authors have investigated the previous studies I think it is still not clear the gap between the previous studies and the current one. Hence, the authors should clarify the gap between the existing research work and the work you intend to do.
6- The structure of this article can be strengthened. Please draw a flow chart of this article and place it at the end of the Section 1 Introduction.
7- The authors have to clearly state the limitations of their study.
8- The following articles can be considered and added to the introduction part of the study to improve the quality of the study. 1) The Fixed-Time Observer-Based Adaptive Tracking Control for Aerial Flexible-Joint Robot with Input Saturation and Output Constraints; 2) BOLD Bio-Inspired Optimized Leader Election for Multiple Drones; 3) Analysis of Wavelet Controller for Robustness in Electronic Differential of Electric Vehicles An Investigation and Numerical Developments; 4) W Band Mini-SAR on Multi Rotor UAV Platform; 5) Design optimization of a fixed-wing aircraft; 6) A Review of Commonality Design for Civil Aircraft; 7) UAV Trajectory Planning for Complex Open Storage Environments Based on an Improved RRT Algorithm.
9- Please add a conclusion part to the manuscript with only the most important findings.
Comments on the Quality of English LanguageWhile the work is well-written in general, it, unfortunately, contains some grammatical and typographical problems. Before resubmitting the manuscript, it is suggested that the authors reread it.
Author Response

(The authors gave the same response as above.)

Reviewer 4 Report
Comments and Suggestions for Authors
1st row – The first half of the article is written more like a review and not like an article. The second half already contains practical examples and analysis.
In the 35th row –“ 2][4. ” incorrectly written source citation.
-why are the sources not cited according to the first occurrence in the text 1, 2, 3, 4,...? Then [2] and [3] should be here.
In the 41st row - when using the abbreviation RMB for the first time, the abbreviation RMB must be explained.
The introduction chapter is very general. On the contrary, information on how exactly the results of the found match will be used in the conclusions. The benefits are mentioned here, but the information about the specific use of the results is missing.
In the 122nd -123rd row - please explain how the weights are determined.
In the 147th -148th rows - please explain what data is entered after X and Y.
In the 1st figure- unify the dimensioning style according to R because the dimensioning at D creates a dummy apostrophe.
In the 4th and 5th equations- how the weights w1 and w2 are determined.
In the 3rd figure, the Inertial navigation system should be INS.
7th equation - could you describe the equation using a practical example?
- please explain how alpha, beta, gamma, and omega values are determined.
In the 5th figure – unify the font size in the image description.
How did you get the data in the 7th Figure? Is it based on Equation 7?
In the 6th figure – unify the font size in the image description.
In the 2nd table – units should be in square brackets (also applies to other tables).
Table 3 is over three pages. It should be appropriately divided into several smaller tables. The first column in the table header is incomplete. You can put this in the name of the table.
Table 4 – inappropriately chosen table name. Please indicate what kind of data it is.
In the 370th row - "civilian aircraft", wouldn't it be better to limit yourself to UAVs or some group so that the results are clearer and, in the case of other groups, just a verbal comment? It's so confusing.
In the 440th row - what are the data sources? Did you get everything from datasheets, or as a reader, will you work on the data/specifications?
In the 445th row - how to fill in the missing data manually. You write that all data were collected manually. If something is missing, how do I add it? Do you mean to calculate from existing data?
In the 455th row - I don't think so. There will be no problem with intellectual property (patent law, utility model and other forms of protection). If we talk about conformity, it is not an imitation rather than a result of development.
In the 11th figure - the text passing through the axis is illegible.
- complete the marking of the horizontal axis
10th table - you should not divide the table into two pages - at least give a name to the second part of the table or move it to a new page
12th table – place the name of the table above the table.
Summary - can you describe how manufacturers have proceeded so far when designing new UAV models?
- Now, are they using your model?
- when using your model, there is no suppression of innovation during development if they focus on conformity
- what about the overvalued parameters of the manufacturers - did you also verify the specification?
- can your model also be used for AGV or other products? Are you thinking about it?
Author Response

(The authors gave the same response as above.)

Round 2
Reviewer 1 Report
Comments and Suggestions for Authors
The author answered my question.
Reviewer 3 Report
Comments and Suggestions for Authors
No further comment